# Advancements in the Application of Nanomedicine in Alzheimer’s Disease: A Therapeutic Perspective

**DOI:** 10.3390/ijms241814044

**Published:** 2023-09-13

**Authors:** Nidhi Puranik, Dhananjay Yadav, Minseok Song

**Affiliations:** Department of Life Sciences, Yeungnam University, Gyeongsan 38541, Republic of Korea; nidhipuranik30@gmail.com (N.P.); dhanyadav16481@gmail.com (D.Y.)

**Keywords:** neurodegenerative disease, Alzheimer’s disease, nanomedicine, blood–brain barrier, therapeutic agent

## Abstract

Alzheimer’s disease (AD) is a progressive neurodegenerative disease that affects most people worldwide. AD is a complex central nervous system disorder. Several drugs have been designed to cure AD, but with low success rates. Because the blood–brain and blood–cerebrospinal fluid barriers are two barriers that protect the central nervous system, their presence has severely restricted the efficacy of many treatments that have been studied for AD diagnosis and/or therapy. The use of nanoparticles for the diagnosis and treatment of AD is the focus of an established and rapidly developing field of nanomedicine. Recent developments in nanomedicine have made it possible to effectively transport drugs to the brain. However, numerous obstacles remain to the successful use of nanomedicines in clinical settings for AD treatment. Furthermore, given the rapid advancement in nanomedicine therapeutics, better outcomes for patients with AD can be anticipated. This article provides an overview of recent developments in nanomedicine using different types of nanoparticles for the management and treatment of AD.

## 1. Introduction

Today, the world’s foremost cause of death and infirmity is central nervous system; (CNS) disorders, mostly involving neurodegenerative diseases (ND), which cause great pain to patients and their relatives. NDs, including Alzheimer’s disease (AD), Parkinson’s disease (PD), epilepsy, dementia, Huntington’s disease, amyotrophic lateral sclerosis, multiple sclerosis (MS), brain stroke, and injuries, are the most common CNS diseases [1,2]. Gustavsson et al. (2022) projected that 416 million people globally fall into the AD continuum, which is significantly higher than the frequently reported estimate of 50 million individuals experiencing dementia [3]. Neurofibrillary tangles (NFT) and amyloid beta (Aβ) plaques are signs of the disease. Although AD has well-known neural characteristics, its symptoms are momentarily alleviated by the available treatments [4]. Many medicines licensed for the treatment of cognitive deficits are based on the manipulation of neurotransmitters or enzymes. However, difficulties related to poor drug solubility, low bioavailability, and inability to overcome barriers along the drug delivery route impede the development of new therapeutic options [5]. Therefore, treatment technologies must overcome the obstacles posed by the blood–brain barrier (BBB). Only a few molecules can pass through this intricate and tightly controlled barrier, which examines the biochemical, physicochemical, and structural characteristics of neighboring molecules at the perimeter. Numerous nanotechnology-based platforms have been developed to improve therapeutic efficacy in the brain [6,7]. These aided medication delivery techniques utilize advanced design principles and have several advantages over conventional techniques [8]. For example, formulations are highly customizable to increase drug loading, targeting, and release efficacy, and nanoparticles (NPs) are often low-cost technologies that can be employed for non-invasive administration [9]. These nanoscale structures make it easier for drugs to cross the BBB, enhancing their pharmacokinetics, pharmacodynamics, and bioavailability. Polymeric NPs, dendrimers, and lipid-based NPs are examples of such nanocarriers [10]. This article aimed to gather relevant information concerning the barriers in drug design for AD and the clinical use of NPs in the development of a new formulation for AD treatment. We searched the PubMed, Google Scholar, and Scopus databases for review articles, research articles, studies on nanomedicine, and clinical trials relevant to the fields of nanomedicine-based treatments against AD. Moreover, this review provides a brief overview of AD, barriers to developing therapeutic agents, the role of NPs, and their clinical aspects. 

## 2. Alzheimer’s Disease 

AD is a severe, fatal, progressive brain illness that impairs cognitive abilities [11]. The two most popular theories for the pathogenesis of AD are the excess and aggregation of Aβ peptide in the form of plaques and microtubule-associated tau protein becoming abnormally hyperphosphorylated and forms NFT that accumulates in the brain of AD; however, synaptic disturbance for calcium influx, autophagy, microgliosis–astrogliosis, and neuron demyelination are considered major factors in AD pathophysiology (Figure 1). Recent research has shown that chronic oxidative stress (OS), mitochondrial dysfunction, hormone imbalance, inflammation, genetic components, or BBB dysfunction may also play crucial roles in the pathogenesis of AD [12,13]. Food and Drug Administration-approved therapies for AD are currently available and are related primarily to two biochemical pathways involving the buildup of Aβ-peptide and NFT of p-tau protein. However, they have poor effectiveness in treating AD, demonstrating the necessity for other therapeutic strategies. Identifying additional molecular targets through discovering novel biomarkers may help the development of new treatments for AD [14].

The emergence of intracellular NFT and the production of extracellular Aβ plaques in the brain are key indicators of AD onset [15,16]. The histopathological characteristics are the loss of hippocampal neurons, synaptic degeneration, and aneuploidy. Additionally, early pathophysiological changes in AD have been linked to neuroinflammation, OS, mitochondrial dysfunction, and an injured brain lymphatic system [17,18]. 

AD is complicated by several variables, including genetic, environmental, health, and lifestyle factors. Obesity, hypertension, high cholesterol, and diabetes are only a few conditions that increase the risk of developing AD [19,20,21]. 

Researchers have studied many therapeutic targets. Among the therapeutic targets, common strategies include Aβ-based, tau protein-based, NFT-based, mitochondrial, OS, anti-inflammatory, and blood–brain receptor targeting (Table 1). 

## 3. Challenges of Drug Designing for AD Treatment

The brain is regarded as the best-protected body organ. Several protective barriers surround the brain, including the skull, meninges, cerebrospinal fluid, and BBB. These elements contribute to the brain’s defense against external and internal harm as well as helping it stay healthy. However, in a diseased state, these protective barriers make it more difficult for therapeutic drugs to reach the brain [58]. The major barriers, including the BBB, blood–cerebrospinal fluid barrier (BCFB), and multidrug resistance proteins (MDRPs), are briefly discussed. 

### 3.1. The Blood–Brain Barrier

The BBB is the principal barrier that prevents nanomedicines from entering the brain, limiting their ability to treat and diagnose AD. Therefore, nanomedicines should be intelligently developed to increase their capacity for brain accumulation [59,60]. The BBB, which prevents many medications from entering the brain at adequate concentrations, is the main obstacle in treating neurological illnesses. Moreover, unintended adverse effects resulting from drug release from the carrier before it reaches the brain further complicate treatment efforts. The BBB comprises endothelial cells, basal membranes, pericytes, and astrocytes (Figure 2). Tight junctions provide unique characteristics to the BBB. Consequently, the BBB prevents the entry of medications that are effective in treating a variety of neurological illnesses.

Brain capillaries appear to have essentially no intercellular interstitial gaps. As a result, transit likely involves all cells. Consequently, lipid-soluble compounds can pass freely through all endothelial membranes and easily cross the BBB [61]. Various different strategies are mentioned in recent studies, including direct injection, nose-to-brain route, the opening of BBB, inhibition of efflux transporters, and use of nanocarrier [62]; however, among these strategies, nanocarrier seems to be the most appropriate and result-oriented. 

Using cutting-edge therapeutic nano-delivery devices may solve the problem of crossing the BBB [63,64,65,66,67]. Because of their distinctive physicochemical features and capacity to penetrate the BBB, engineered NPs < 100 nm have multiple applications in resolving these biological and pharmacological problems [68]. The ability of NP to cross the BBB increases the likelihood of early diagnosis and successful treatment of neurological diseases [69]. However, six different access routes that could be used to cross the BBB are paracellular diffusion (hydrophilic molecule), transcellular diffusion, carrier-mediated transport, receptor-mediated transport, adsorptive-mediated transport, and cell-mediated transport [62]. Paracellular transport is a passive transport process that results in the transport of substances across an epithelium. Tight junctions are the major rate-limiting pathway in the paracellular transport of large molecules across the epithelium [70]. Large molecules such as polypeptides are generally excluded from paracellular diffusion due to their hydrophilicity and high molecular mass. Adsorption-mediated transcytosis has attracted much attention because of its potential for large-molecule drug delivery to the brain. Cationic albumin-mediated brain DDS is commonly used; however, coupling this component with NP DDS is proven to be successful in cross-BBB for CNS delivery. Polymeric NPs are commonly used for adsorption-mediated transcytosis [71]. Immunocytes, neutrophils, and lymphocytes have a high degree of mobility, and they can cross various barriers to reach the target site and release their medication cargo. To enhance drug delivery, these cells can be loaded with nanocarriers such as polymeric NPs. In receptor-mediated DDS, transferrin receptors, low-density lipoprotein receptors, and insulin receptors are commonly used. A complex is formed between the desired drug and a receptor-targeting agent. This component is then linked or incorporated into the NPs [72]. Transcellular diffusion of molecules is based on their solubility, their molecular mass, and charge [73]. One important process for drug transportation over the BBB carrier-mediated transcytosis is used for the entrance of nutrition and energy intake into the brain cells. The benefits of carrier-mediated transport have been grafted onto nano DDSs using a technique that involves functionalizing the surface of the tiny molecules with certain ligands [74]. Among these six pathways, except the cell-mediated pathway, other five pathways are commonly used in nanomedicine delivery that cross the BBB for AD management, as shown in Figure 3. 

### 3.2. The Blood–Cerebrospinal Fluid Barrier

The BCFB, which is constructed by epithelial cells of the choroid plexus and systematically deals with drug compounds, is the next barrier to overcome after the BBB. The choroid plexus epithelium (CPE) is one of the most effective tissue types and is based on secretory systems that balance cellular transport mechanisms. This epithelial barrier divides the blood and CSF in a manner similar to that of the BBB. Cells near the CSF-facing surface had an adjacent CPE that was tightly preserved. The CPE comprises detoxifying enzymes and multi-specific efflux transport proteins that work together to block the entry of potentially deadly substances into the CNS. Several medication delivery strategies have been developed; most are suited to the BBB and target the microvascular endothelium [75,76]. For the treatment of CNS disorders, targeting the BCFB, which is sculpted by the choroid epithelium, is crucial, in addition to targeting the BBB [77].

In order to enhance the administration of CNS therapies and offer brain protection measures, the BCFB, a less well-studied brain barrier location than the BBB, may serve as a possible therapeutic target. Therefore, by conducting a preliminary evaluation of the anticipated physiochemical behavior of a drug against this barrier, the use of reliable and accurate in vitro models of the BCFB can reduce the time and effort spent on pointless or repetitive research efforts. The development of innovative CNS treatments and pharmacologically active candidates that take advantage of the choroid plexus epithelial cells’ inherent transporting ability will be beneficial for developing CNS drug delivery strategies through the BCFB. There are still questions concerning the CP’s suitability for this purpose despite the fact that it has not received much attention as a prospective route for medication delivery to the brain. By using the current in vitro BCFB models, preliminary mechanistic and transport experiments can be used to assess the efficiency of the design process and the rate of successful transport across this barrier. Protein receptors, solute carriers, and amino acid transporters are three key groups of possible transporters that could be targeted to facilitate medication delivery to the CNS across the BCFB [78].

### 3.3. Multidrug Resistance Proteins

The BBB is formed by endothelial cells within brain capillaries and this cerebral endothelial cells contain diverse metabolic enzymes, including cytochrome P450 enzymes, alkaline phosphatases, and glutathione transferases, as well as energy-dependent efflux transport proteins, such as P-glycoprotein (P-gp), breast cancer resistance protein (BCRP; ABCG2), and MDRPs, which are arranged asymmetrically and act as barrier [79,80,81]. These MDR transporters, P-gp and BCRP, belong to the ATP-binding cassette (ABC) superfamily and prevent exposure of the brain to a large range of molecules [82]; this makes therapy unsuccessful by preventing some drugs from precisely achieving their intended therapeutic goals while also reducing drug buildup in the brain [83]. Among these transporters, MDRP is a main barrier for drug distribution to the brain due to its concentrative efflux activity. Changes to the expression levels of MDRP possibly alter drug concentrations in brain and spinal cord tissues. An alternative approach could be used for drug delivery by inhibiting the expression and function of P-gp [84]. Therefore, one of the current key issues in pharmaceutical research is the development of tools that allow for the efficient and effective delivery of medications into the CNS by the MDRP system [85]. Simultaneously, there is a need to gain a better understanding of ABC transporter activity at the BBB and BCFB that could correctly predict the drug pharmacokinetics and pharmacodynamics in the CNS. 

The major feature of AD, a multifactorial ND that primarily affects older people, is a progressive decline in cognition, emotion, language, and memory. Pharmaceutical medication formulations and cholinesterase inhibitors have been widely promoted as AD treatments because of their beneficial effects. To enhance patients’ cognitive results, these drugs were eventually found to be unsuccessful in addressing the underlying causes of AD pathogenesis and instead focused only on symptoms [86,87].

Therefore, the market’s current drug delivery technologies cannot offer sufficient cytoarchitectural recovery and interconnections, which are essential for real recovery in AD. The concepts and methods of nanotechnology mentioned in this review can also substantially eradicate AD. However, nanotechnology can overcome these limitations by developing novel carrier-based platforms focusing on the targeted selective release of pharmacological payloads with on-demand and controlled release kinetics and improved reach by altering or evading the BBB. Nanomaterials have been studied for administering anti-AD drugs in experimental models of AD. This review covers several synergistic strategies in depth, providing details on the symptoms, risk factors, treatment options, and function of nanocomposites in slowing the progression of AD [88].

## 4. Nanomedicine

Maintaining the bioavailability, pharmacodynamics, and pharmacokinetics of medications is crucial to exerting their full therapeutic effects. Therefore, the goal of integrating a drug into or onto a polymeric and/or lipid NP is to substantially increase the drug’s therapeutic effect. The use of NPs in drug delivery is advantageous because they increase a drug’s bioavailability by enhancing its aqueous solubility and lengthening its half-life, which, in turn, slows the pace of drug clearance and delivers the medication to its intended site of action [89,90,91]. Furthermore, a wide range of NPs is used to treat AD [92]. Figure 4 shows the use of various NPs in the development of nanomedicines for the management and treatment of AD.

Nanoliposome carriers are the most promising drug delivery methods because of their biocompatibility and high flexibility in delivering a variety of therapeutic compounds across the BBB and specifically targeting brain cells. Evidence demonstrates that PSLs can cross the BBB to enter the brain, where they interact with several types of neuronal cell receptors [93]. 

### 4.1. Metallic/Inorganic Nanoparticles 

Metallic NPs have been extensively used in recent years for the diagnosis, imaging, and treatment of NDs. Metallic NPs commonly contain gold NPs (AuNP), silver (Ag) NPs, iron oxide (FeO) NPs, platinum (Pt) NPs, and ceria (Ce_2_O_3_) NPs. These NPs play a significant role as DDSs in brain-targeted therapies. Metallic NPs can be easily fabricated and functionalized by adding different functional groups, and their shapes and sizes can be modified [94]. The conjugation of metallic NPs with drug molecules, antibodies, DNA/RNA, peptides, and aptamers has been extensively studied in recent years [95]. Among the metallic NPs, AuNPs have been widely used in diagnosis and imaging, such as in DDS. 

Because of their biocompatibility, fascinating optical features, surface functionalization, and non-immunological qualities, AuNPs have gained significant interest as novel platforms for catalysis, drug transport, and disease diagnosis and treatment. AuNPs have also been developed to treat various CNS illnesses owing to their various dimensions, forms, and surface features [96]. Numerous researchers have used AuNPs to develop therapeutic compounds for the treatment of AD. According to Shao et al. (2023), AuNPs have been used to modify Aβ fibrillation associated with AD in intracellular and extracellular areas [97]. An aptamer-conjugated polydopamine-coated AuNP formulation was prepared to treat AD and tested on PC12 cells. The formulation inhibits the aggregation of the Aβ peptides and prevents Aβ-induced cell membrane damage [98]. In 2015, Gao et al. designed an AuNPs-based formulation where AuNPs were conjugated with polyoxometalate as an Aβ inhibitor for AD treatment. The formulation effectively inhibits Aβ aggregation, dissociates Aβ fibrils, and reduces peroxidase activity and cytotoxicity. This success is attributed to the advantageous use of AuNPs as BBB-crossing carriers [99]. In a recent study, AD model rats were treated with AuNPs conjugated with okadaic acid (OA). AuNPs were observed to maintain p-tau levels in the cortex and hippocampus; however, OA increased p-tau levels. Rats treated simultaneously with OA and AuNP showed high interleukin (IL)-1β and IL-4 levels in the hippocampus and cortex. AuNPs exhibit antioxidant properties and reduce OS in the brain [100]. Sanati et al. studied the impact of AuNPs on Aβ-induced AD in experimental rats, finding that AuNPs may enhance learning and memory for spatial information, and the expression level of stromal interaction molecules was increased, leading to an improvement in neural survival [101]. Studies have shown that AuNPs are multifunctional therapeutic agents for the treatment of AD.

Selenium nanoparticles (SeNPs) have received considerable attention because of their exceptionally low toxicity and antioxidative qualities. Selenium is a micronutrient crucial for preserving human health and lowering OS in the brain. Because of their outstanding neuroprotective properties, they have been implicated in the prevention of AD [102]. Selenium- and sodium selenite-rich NPs have antioxidant qualities and are utilized to treat brain illnesses; they also enhance cognitive function in a size-dependent manner [103]. Yin et al. conducted in vitro studies to examine the interactions between B6-SA-SeNPs and amyloid peptides as well as to assess the effectiveness of B6-SA-SeNPs when delivered to brain capillary endothelial cells. Because of their anti-amyloid, antioxidant, and high brain transport efficiency, the findings indicated that B6-SA-SeNPs could be a useful treatment for AD and a novel nanomedicine for disease modification in AD [102]. 

Neurotoxicity, DNA damage by inorganic NPs, and biochemical properties of the brain and iono-regulatory processes are the major challenges of this nanocarrier in AD therapeutics. 

### 4.2. Carbon-Based Nanoparticles

A large category of carbon-based nanomaterials, known as “carbon dots” (CDs), are broadly divided into carbon nanodots, polymer dots, carbon nitride dots, and graphene quantum dots (QDs), which are used as DDSs for the treatment of NDs [104]. Top–down and bottom–up strategies were used to synthesize CDs, and a range of reaction conditions and precursors were used. The successful penetration of the BBB by a variety of CDs and CD ligand conjugates indicates encouraging advancement in the use of CD-based DDS for the treatment of CNS disorders [105].

The development of CDs has resulted in numerous advancements in recent years, including the delivery of medications that can traverse the BBB [106,107,108]. CDs are biocompatible and harmless because they do not contain metals. Owing to their abundance of surface functional groups, CDs can be conjugated to a variety of pharmacological compounds as nanocarriers through either covalent or noncovalent interactions. Most importantly, the combination of substantial surface diversity and small particle size simplifies the transfer of drugs across the BBB via active pathways. Additionally, the photoluminescence of CDs makes it simple to follow their in vivo movements in real time. With the help of these promising NPs, additional medications can now be transported to the brain, rendering the fight against AD and other NDs a new weapon [109]. One published study demonstrated the significance of yellow-emissive CD (Y-CDs) in AD treatment. According to these results, Y-CD crosses the BBB, enters cells, and inhibits the overexpression of human amyloid precursor protein and Aβ [106]. Another study also revealed the potential of CDs as a multifunctional β-sheet breaker and provided a promising anti-Aβ aggregation strategy for AD treatments [110]. In 2022, Yan et al. developed multifunctional CD photosensitizer nanoassemblies to inhibit amyloid aggregates that could easily cross the BBB and prevent microbial growth after treatment [111]. These recent studies demonstrate the potential significance of CD in CNS and its promising role as an inhibiting agent for the Aβ-related pathology of AD. Chung et al. fabricated targeting CDs that were red light-responsive and capable of suppressing Aβ aggregation and neurotoxicity in the brain of a patient with AD [112]. 

QDs are NPs with electronic and optical characteristics such as fluorescence and strong light emission. They possess distinct qualities, such as photostability, high quantum yield, high emission, and size tunability. The use of theranostic NPs in sensing, imaging, and medication delivery is currently receiving considerable attention. Additionally, QDs hold significant promise for the identification and management of several CNS disorders, including PD, AD, and MS. In several areas of life sciences, QDs are semiconductor nanocrystals with unique optical features, including high photochemical stability, extended wavelength, and long fluorescent half-life [113]. Some varieties of conjugated QDs are suitable carriers and are valuable for traversing the BBB in a variety of applications, including accurate imaging and detection of brain illnesses. The primary factor for transferring across the BBB is the particle surface, not the NPs’ size or chemo-electric charge [114]. One study synthesized QDs from the extract of the flower of *Clitoria ternatea* and observed that the learning and memory capacity of treated rats improved in the radial arm and Morris water maze assays. These results also confirmed that QDs significantly inhibited AChE, increased the levels of glutathione and proteins, and decreased the levels of lipid peroxide and nitric oxide [115]. In a study by Wang et al., a nanoformulation of graphene oxide (GO)-loaded dauricine was developed for the treatment of AD and was evaluated in an AD experimental model both in vivo and in vitro. The findings demonstrated that GO loaded with dauricine significantly decreased OS by increasing superoxide dismutase levels, lowering reactive oxygen species, and decreasing malondialdehyde levels in vitro. It also significantly improved cognitive memory impairment and brain glial cell activation in mice with Ab1-42-induced AD. Thus, GO-loaded dauricine has the potential to be a useful drug for the rapid treatment of AD. This study showed that GO loaded with dauricine protected against Ab1-42-induced oxidative damage and apoptosis in both AD models [116].

A recent study investigated the role of graphene oxide as a chelator to dissolve the amyloid-β plaques in AD using the density functional theory study. They have determined the chelating ability of various metals and GO by comparing the binding energies between metal—chelator and metal–Aβ complexes and conclude that GO can be used in future AD drug delivery therapy to target toxic metal–Aβ interactions and reduce Aβ aggregation [117].

GO may block the mTOR signaling pathway and initiate autophagy in microglia and neurons. When microglia and neurons were co-cultured, GO triggered autophagy in both cell types, particularly in the microglia, which aided in the removal of Aβ and ultimately had a protective impact on the neurons. Additionally, GO was able to decrease the toxicity of Aβ to neurons by its clearance in addition to being non-cytotoxic to microglia and neurons. These findings demonstrate the possibility of GO as an AD treatment [118]. 

Carbon nanotubes (CNTs) have garnered attention for their unique properties and biomedical potential. The ability of high drug loading, along with the capability to readily cross the BBB and also facilitate the passage of drugs to the brain via the olfactory route with ease, opens up new possibilities for targeted drug delivery and neuroprotection. In recent research, β-cyclodextrin modified CNTs nanocomposite was prepared for delivery of donepezil hydrochloride drug for the treatment of AD and found the significant sensitivity and limit of detection [119]. 

Since their unique characteristics, such as their ability to easily pass through cell membranes, thermal properties, large surface areas, and ease of molecular modification, carbon NPs stand out as highly innovative materials that could be used in novel therapeutic regimens against CNS. However, their usage in humans is also constrained, mostly due to the beginning of toxic phenomena that impact the nerve cells and the onset of inflammatory and oxidative processes. The focus of the new research must now be on the hunt for novel, more useful target molecules while also using appropriate engineering to structure nanomaterials that are better suited for human use [120].

### 4.3. Lipid-Based Nanocarriers

The abovementioned constraints are addressed using nanotechnology-based methods that utilize liposomes, micelles, dendrimers, and solid lipid nanoparticles (SLNPs) as DDSs. This study focused on medication delivery to treat NDs using an SLNP formulation [121]. Notably, SLNP can pass physiological barriers to increase bioavailability without using high-dosage forms, direct the active compound toward the target site with a significant reduction in toxicity to the adjacent tissues, and protect drugs from chemical and enzymatic degradation. We believe that SLNP may be an effective method for medications to cross the BBB and reach damaged parts of the CNS in patients with NDs such as AD and PD [122]. One study showed that nicotinamide-loaded functionalized SLNPs improved cognition in AD animal models by reducing the tau hyperphosphorylation [123]. Rivastigmine tartrate-loaded SLNPs have been formulated for enhanced intranasal delivery to the brain in the Alzheimer’s therapeutics [124]. In a therapeutic study on AD treatment, RVG29-functionalized SLNPs were prepared for the delivery of quercetin to the brain, resulting in a 1.5-fold increase in drug permeability to the experimental cell line [125]. 

Small vehicles (≤100 nm) are frequently preferred because only specific sizes can pass through the BBB in AD. However, studies have demonstrated that liposomes between 100 and 140 nm offer several benefits, including a longer half-life in blood circulation and a lack of plasma proteins [126]. In a recent study, transferrin-functionalized liposomes conjugated with vitamin B12 were used as therapeutic agents for AD. The formulation targeted the BBB and neuronal cells and delayed the formation of Aβ fibrils, showing great potential for AD therapeutics [127]. Caffeic acid-loaded transferrin-functionalized liposome NPs were developed capable of preventing AD by blocking Aβ aggregation and fibril formation and disaggregating mature fibrils [128]. 

Numerous studies have demonstrated the effectiveness of dendrimers as DDSs for the treatment of various diseases [129,130]. Since positively charged NPs, in conjunction with mucus, have greater cellular absorption, as described in a previous study [131], superficially positively charged dendrimers are thought to facilitate enhanced drug transport to the brain. A recent study revealed that polyamidoamine (PAMAM) dendrimers are suitable for the delivery of donepezil, an AChE inhibitor used to treat AD, have good in vivo pharmacokinetics and can cross the BBB [132]. Tacrine and PAMAM dendrimers co-delivered as therapeutic molecules for AD treatment were observed to help reduce the side effects of the drug in an animal model [133]. In the ND therapeutic study, carbamazepine was co-administered with PAMAM dendrimers in zebrafish larvae, resulting in reduced side effects from the carbamazepine [134]. 

Nanoliposomes are a feasible and promising DDS for AD that has not yet been tested in clinical trials. They are highly biocompatible and can carry various therapeutic molecules across the BBB and brain cells. They can be tailored to extend blood circulation time and directed against individual or multiple pathological targets.

In comparison to equivalent free drug solutions/suspensions, nano-lipid drug delivery systems have been shown to have positive properties such as prolonged drug release, increased CNS bioavailability, and further improved therapeutic efficacy [135]. 

### 4.4. Polymeric Nanoparticles 

Polymeric NPs are solid objects composed of organic colloidal NPs with polymetric, natural, or artificial materials. Numerous monomers can be used to create polymeric NPs, which can then be polymerized using various methods, and their characteristics can be tailored for diverse applications. Synthetic, natural, and hybrid polymeric nanoparticle systems are most frequently used for brain targeting [136]. Chitosan and PLGA are the most common polymeric NPs used in AD treatment. Mathew et al. conducted an in vitro study of PLGA-coated curcumin with Tet-1 in AD treatment, finding that PLGA-coated curcumin NPs have the potential for AD treatment because of their anti-amyloid and antioxidant properties [137]. Another study showed that PLGA NPs loaded with Aβ generation inhibitor S1 and curcumin remarkably decreased the level of Aβ, enhanced the activities of antioxidant enzymes, and attenuated memory deficits and neuropathology in AD mice [138]. A recent study found that a formulation of PEGylated biodegradable poly (lactic-co-glycolic) nanospheres loaded with dexibuprofen administered to APPswe/PS1dE9 mice reduced memory impairment and decreased brain inflammation more efficiently than the free drug [139].

The size, shape, and chemical and physical properties of the NPs included in this review were not uniform. Each nanoparticle has unique properties with many advantages and disadvantages. The advantages and limitations of these NPs as potential DDSs have been discussed in many articles; however, scientists and researchers are working on most nanomaterials to study their uniformity, drug loading, drug release capacity, and safety as DDS for treating AD and other neurological diseases. Several in vitro and in vivo studies have demonstrated the significance of different types of NPs for treating AD. A few of these are discussed in Table 2, along with their results and possible prospects or limitations/advantages. 

Clinical failures might be brought on by polymeric nanoparticles’ potential toxicity and sluggish degradability. The capacity of the polymeric NPs to traverse the BBB is constrained without surface changes. Although these NPs have a good rate of disintegration, the chronic nature of neurodegenerative illnesses and the requirement for repeated doses of the therapeutic substance could result in polymer accumulation that could have unwanted side effects [78,175]. 

## 5. Discussion

Mental and CNS illnesses have a significant prevalence worldwide, including neuroinflammation, brain tumors, and NDs. Neuronal loss is the main characteristic of NDs. AD and PD are the most common NDs. Although several medications are now approved for managing NDs, most treat only the related symptoms [176]. Because of the presence of the BBB, developing therapeutic agents for various CNS-related disorders is highly challenging [177]. 

The BBB serves as the primary barrier to drug delivery in AD treatment. The physical and biochemical barriers of the BBB limit the effects of all hydrophilic drugs; efflux pumps inside the BBB frequently transfer lipophilic drugs back into the blood [178]. As a result, it is critical to understand the structure and activity of the BBB to suggest alternative methods for delivering AD medications via nano-DDS. Utilizing the endothelial cell-binding affinity of lipid-soluble NPs can increase the rate at which a drug is transported via endocytosis or lipophilic transcellular routes. Moreover, the adsorptive properties of NPs can be useful because they can adhere to blood capillaries in the BBB, boosting the likelihood that the target medicine will cross this barrier [179,180]. Altering NPs with specific receptors, carrier proteins, and receptor-mediated transcytosis can enhance medication uptake through the BBB. Despite the ability of NPs to pierce the BBB, only approximately 5% of medications reach the brain, leaving the remaining 95% at an ineffective location [181]. This phenomenon may cause systemic side effects because the conventional route of drug administration is ineffective for accurately delivering the agent to the brain. The transport of therapeutic substances to the CNS via the olfactory and trigeminal nerves of the nasal cavity is facilitated by intranasal administration.

Furthermore, intranasal administration is risk-free and non-invasive. This medicine can prevent hepatic first-pass metabolism and drug degradation, increasing drug bioavailability [182]. Nanotechnology is a cutting-edge method that may open the door to new ways to overcome blood–CNS barriers, particularly the BBB. The therapeutic effects of drug delivery systems depend on their capacity to bypass the immune system, pass through the BBB, and locate themselves in the target tissues [1]. NPs are widely used in the diagnosis and treatment of AD [183]; however, for successful results, several aspects must be considered, including the target-specific distribution and metabolism of NPs and the interactions between target molecules (Aβ/Tau/NFT) and NPs. Nanomaterials have good in vitro inhibitory efficiency; however, the in vivo inhibition efficiency needs to be increased [184]. Several metallic NPs provide a good platform for efficient inhibition of the aggregation of Aβ either directly or after conjugation with a drug molecule [185]. 

Traditional DDSs are ineffective because they cannot pass through the BBB. Although innovation in research ensures the creation of nanotheranostics, a major issue is its potential for use in human therapies. The BBB can be crossed by a number of NPs [186]. Researchers are working on the development of nanomedicine using different types of nanocarriers. Table 2 summarizes some of the in/ex vivo studies. Nanoliposome carriers are the most promising drug delivery methods because of their biocompatibility and high flexibility in delivering a variety of therapeutic compounds across the BBB and specifically targeting brain cells. Evidence demonstrates that PSLs can cross the BBB to enter the brain, where they interact with several types of neuronal cell receptors [93].

It is possible to achieve widespread drug delivery to the entire brain by opening the BCFB instead of the BBB because the BCFB is relatively leaky compared to the BBB; water-soluble substances that do not cross the BBB do so on the BCSF barrier and enter CSF at a rate inversely related to molecular weight [187]. Kung Y et al. (2022) effectively validated a strategy for noninvasively and selectively opening the BCFB to promote medication administration into CSF circulation by facilitating drug delivery in the CNS by opening the BCFB barrier with a single low-energy shockwave pulse [188]. 

A promising prospect for the L-type amino acid transporter 1 (LAT1)-mediated brain delivery of CNS medicines is provided by the relatively high expression of the LAT1, which is primarily expressed in the cerebral cortex, blood–brain barrier, and blood–retina barrier. The transporter has proven to be effective in delivering clinically used CNS medications and prodrugs like L-Dopa. The bulk of the time, a parent medication has been conjugated to an amino acid side chain using a biodegradable linker without substituting carboxyl and amino groups in order to effectively bind LAT1 [189].

An advanced method for combating MDR in cancer treatment is small interfering RNA (siRNAs) delivered by nanocarriers. This method increases the effectiveness of already available medications by inhibiting the no-flux and efflux-related protein pumps that are engaged in or overexpressed in the MDR [190,191]. This type of nanocarrier-based siRNA technology may be helpful for treating CNS conditions. Moreover, the downregulation of the pump involved in drug resistance, combining monotherapy with P-gp inhibitors, and bypassing of pumps related to drug efflux are common nanotechnology methods that could be used to overcome the MDR problem in AD treatment.

The rapid development of nanotechnology has made many promising nanomaterials available for biomedical applications. These materials are widely used in the treatment of ND and appear to compensate for some of the shortcomings of stem cell therapy, including the transportation of stem cells, genes, and drugs, control of stem cell differentiation, and real-time monitoring. Therefore, stem cell therapy combined with nanotherapeutic techniques is a promising therapeutic method for treating NDs [192]. Because nanomedicine has been a research area for approximately 20 years, discussing what the field should do in the next 20 years is appropriate for creating a framework based on science for ongoing, sustained progress [193]. 

Nanomedication particles have good stability, are less toxic, and cause fewer side effects when used to diagnose and treat AD in elderly patients. These particles also help patients live more comfortably and have favorable clinical outcomes [142].

Therefore, future research on NP-based brain delivery should concentrate on enhancing safety, precision targetability, and pharmacokinetic characteristics. Human clinical trials should be conducted to assess the efficacy and safety of certain NPs and to determine the most promising and affordable AD treatments. To ensure the safe use of NPs by patients and healthcare personnel, thorough handling and administration standards must be devised. In addition, for the future treatment of AD, the theranostic development of NPs is essential, promising substantial advancements in precision and customized therapy. The clinical outcomes of each patient with AD who received one of these tailored medicines greatly improved. They strongly emphasize evaluating the efficacy and safety of the treatment processes. Thus, NP-DDSs offer strong therapeutic promise for AD. Further translational research should be conducted to examine the interaction of NPs in a biological interface in light of the benefits of crossing the BBB and therapeutic modalities, including symptomatic alleviation, DMT therapy, and hallmark diagnostics. Future alternatives for treating and diagnosing AD are promising when NPs are combined with recently discovered medicines [194]. 

However, the number of patents for nanotechnology-based goods is increasing. Clinical trials are required to assess the therapeutic efficacy and potential toxicological effects on human health. Neurologists and patients will soon benefit from appropriate nanotechnology-based DDSs that may improve therapeutic outcomes at lower costs. Although no clinical studies have been conducted on using nanotechnology to treat AD, this field of medicine is projected to change, bringing new ways to diagnose AD and the ability to individualize a patient’s therapy profile [195].

## 6. Current Research and Future Directions of Nanomedicine

AD is one of the most devastating diseases of the CNS. The primary pharmaceutical obstacle to the effective treatment of AD is BBB and BCFB. BBB-targeted nanomedicine has recently been developed and is now given more consideration in treating AD. Numerous nanoformulations, as indicated in Table 2, have proven to be highly effective in laboratory and clinical investigations. However, clinical translation from bench to bedside is not particularly successful because of an inadequate understanding of the mechanisms of action, the fate of nanocarriers inside the body, systemic toxicity, biocompatibility, aggregation, and the quick clearance resulting from their nanometer size. Comprehensive and in-depth toxicological research should be focused on brain-targeting nanoformulations. The therapeutic potential of nanomedicine will depend heavily on the rational approach and ongoing design of nanomaterials based on in-depth and extensive knowledge of data gathered from biological processes. Polymeric NPs have shown potential toxicity, drug loading into lipid NPs for hydrophilic compounds is weak, and the in vivo stability of liposomes is subpar. Nano-emulsions exhibit poor stability during storage, which causes phase separation and a quick release effect. Dendrimers have toxicity problems as well. To optimize all of these concerns and handle them in the near future, additional in-depth research investigations are, therefore, necessary. 

The major challenges for the development of nanomedicine for AD are shown in Figure 5; however, some critical limitations/challenges are as follows:Biodegradable nature of nanocarrier;Different kinds of functional groups;Different research study has different protocols and show different biodistribution of nanomedicine at different times;The ability of nanomedicine to control its morphological as well as chemical properties in the bloodstream/stability in blood;Nanomaterial does not show aggregation and is non-toxic, target-specific along with being pharmacodynamic and pharmacokinetic;In vivo condition;Reproducibility, predictability, accessibility, and cost-effectiveness;Should cross BBB and another barrier of CNS.

There are a few important things to consider regarding the nanomaterials utilized in the formation of nanomedicine: they are environmentally friendly; cross the BBB; non-toxic by nature; effective; and safe. In addition, nanomedicine delivery must be simple and non-intrusive.

## 7. Conclusions

The tremendous efforts currently needed to treat AD revolve around delivering the correct pharmaceuticals into the right neurons and obtaining enough of the right treatments for the brain parenchyma. However, many unanswered questions and gaps remain regarding the use of NPs for the treatment of neurodisorders, even though many studies have concentrated on the application of NPs in DDDs to overcome the obstacles faced by traditional medicine. Nanocomposites, including metal/inorganic NPs, CDs, QDs, and lipid NPs, have been used to treat AD and other CNS disorders. However, the nanomaterial itself may be cytotoxic and facilitate the passage of neurotoxic compounds across the BBB. Some administration methods may be more invasive or even cause a breach of the BBB, favoring the entry of infections or toxic substances into the brain. Therefore, studying the limitations of these NPs is necessary. 

The most recent developments in this promising subject discussed in this review article can offer systematic knowledge for designing new initiatives and improving the use of NPs to treat AD.

## Figures and Tables

**Figure 1 ijms-24-14044-f001:**
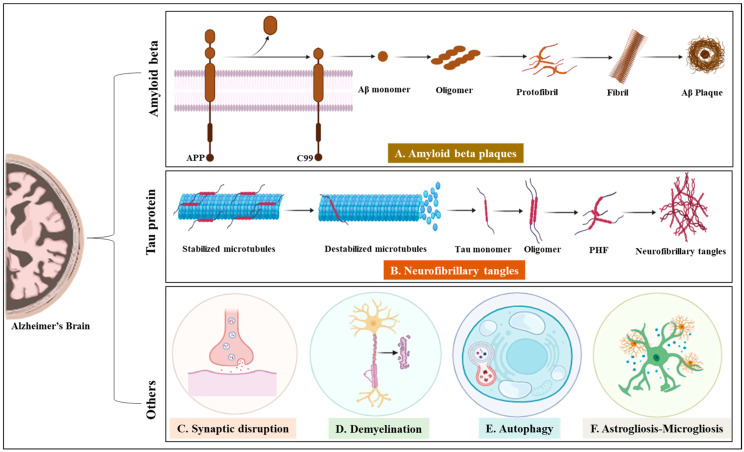
Pathophysiological changes in the CNS in AD. (**A**) Formation of Aβ plaques. (**B**) Formation of NFT. (**C**) Synaptic disruption. (**D**) Demyelination. (**E**) Autophagy. (**F**) Astrogliosis–Microgliosis. (created with BioRender.com accessed on 10 August 2023).

**Figure 2 ijms-24-14044-f002:**
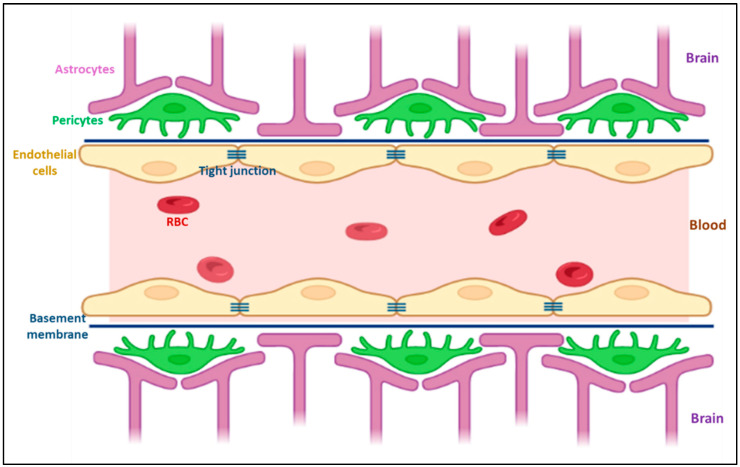
A schematic representation of the blood–brain barrier (created with BioRender.com accessed on 10 August 2023).

**Figure 3 ijms-24-14044-f003:**
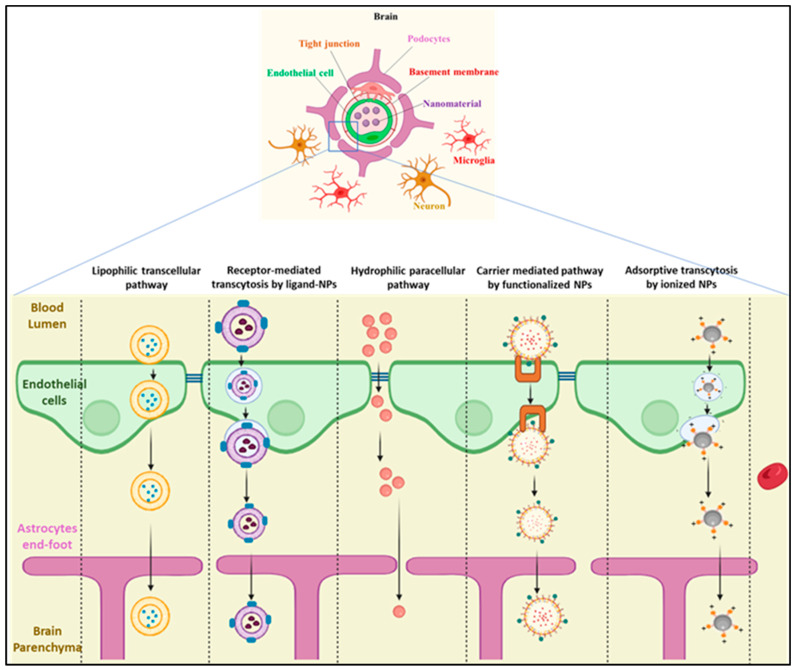
A graphical illustration of possible pathways of nanomaterial-based DDSs that effectively cross the blood–brain barrier for treating Alzheimer’s disease (created with BioRender.com accessed on 10 August 2023).

**Figure 4 ijms-24-14044-f004:**
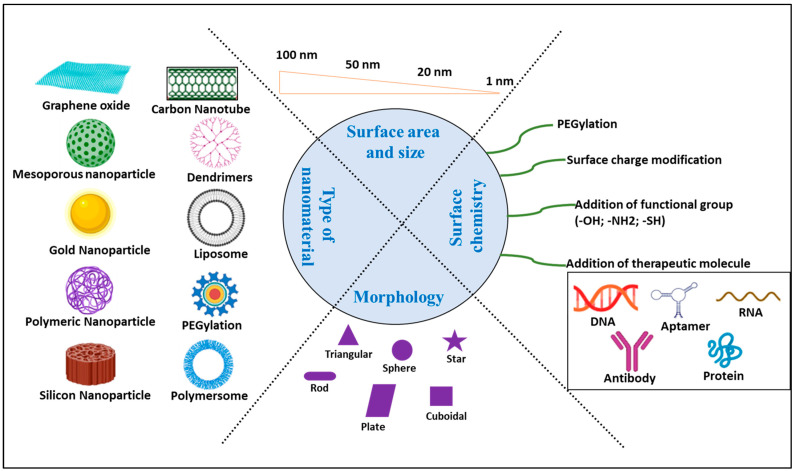
A general overview of nanoparticles used as drug delivery carriers in the management and treatment of AD (created with BioRender.com accessed on 10 August 2023).

**Figure 5 ijms-24-14044-f005:**
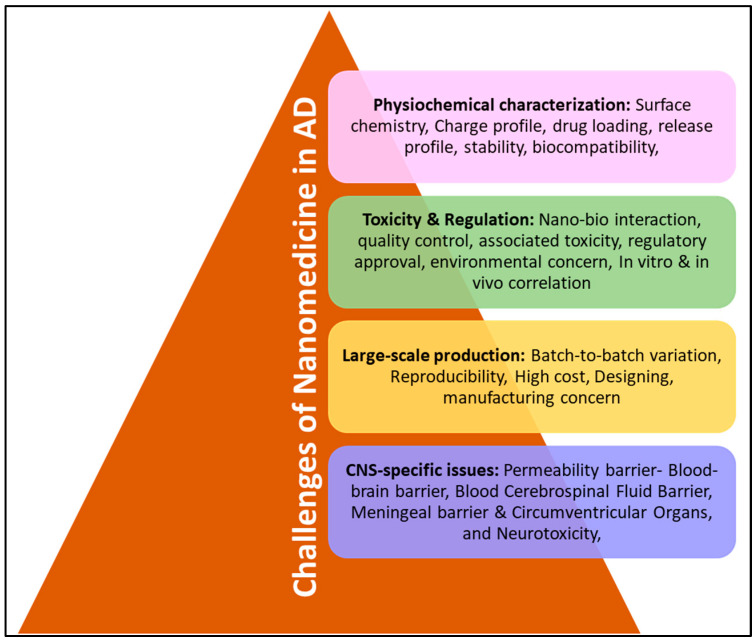
Challenges of nanomedicine for Alzheimer’s therapeutics.

**Table 1 ijms-24-14044-t001:** Therapeutic targets for management and treatment of AD (Individual image was adopted from BioRender.com; 10 August 2023).

Target and Its Structure	Role in AD	Therapeutic Strategies	Common Drugs	References
Amyloid beta protein 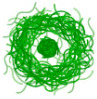	Increases in Aβ concentration, forms oligomers and leads to neurotoxicity	Aβ-protein-targeted (Aβ oligomers treatment) therapeutic strategies for treating AD	Aducanumab, CPO-Aβ17–21 peptide, Huperzine A	[22,23,24,25]
Tau protein 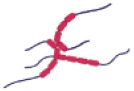	The misfolded Tau is self-assembled, forming a Tau oligomer, and then, Tau inclusions lead to AD	Reduction in toxic Tau gain of function may be an effective therapeutic strategy for AD	Gosuranemab, lonafarnib tilavonemab, semorinemab, zagotenemab,	[26,27]
Neurofibrillary tangles 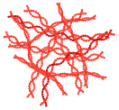	Excessive Tau phosphorylation and aggregation are the key processes in the formation of NFT that cause neuron death and lead to AD	Prevent phosphatase activity; promote microtubule stability; reduce Tau aggregate and formation of NFT	Memantine, AADvac1, ACI-35; Epithilone D	[28,29]
Mitochondria 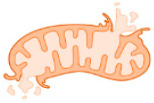	Mitochondrial dysfunction cause AD, which involves OS-induced respiratory chain dysfunction, loss of mitochondrial biogenesis, and mitochondrial DNA mutations	Therapeutics based on antioxidants, anti-apoptotic agents, and molecules that enhance glucose metabolism and mitochondrial bioenergetics	Kaempferol (flavonoid) and rhapontigenin (stilbenoid), P110 and mdivi-1, selenium, SkQ1, MitoApo, astaxanthin	[30,31,32,33]
Oxidative stress 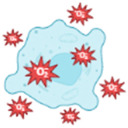	Neuronal cell abnormalities; apoptosis of neurons, cognitive dysfunction that leads to dementia	Antioxidant drug therapy has been investigated as a potential AD treatment	Vitamin C, Vitamin E, -lipoic acid, CoQ10, Curcumin; coenzyme Q and glutathione; melatonin, α-lipoic acid (LA), N-Acetyl-cysteine (NAC)	[34,35,36,37,38]
Neuroinflammation 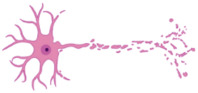	Neuroinflammation leading to memory loss and cognitive decline	Neuroprotective therapeutics using nonsteroidal anti-inflammatory drugs	Pterostilbene, Sulforaphane, Artemisinin	[39,40,41]
Angiotensin receptors 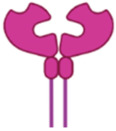	Their role in blood pressure regulation and hypertension leads to neural injury, neuroinflammation, and cognitive function	Therapeutic agents that inhibit RAS, Angiotensin-converting enzymes, ARBs, and block Angiotensin Receptor	Ramipril (ACE inhibitor), ARBs, ACE-Is, RAS-Ms, telmisartan, candesartan, valsartan	[42,43,44,45,46,47]
Secretase receptors 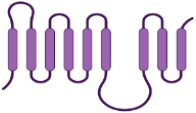	Caused AD by changes in Aβ stability and aggregation	Therapeutic agents that cleaved APP α-secretase have potential to treat AD	MK-8931 (inhibitor of β-site APP cleaving enzyme 1), PRX-03,140 (α-Secretase inhibitors), Begacestat (γ-Secretase inhibitors), CTS-2166 (β-Secretase inhibitors)	[48,49,50,51]
Blood–brain barrier 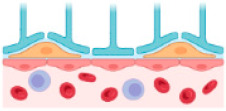	BBB characterizes a link between ND, vascular damage, and neuro-inflammation	Drugs that can penetrate the BBB	ApoE-modifying agents; Neprilysin	[52,53,54]
Cholinergic insufficiency 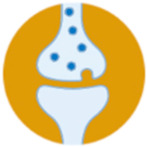	Reduced choline acetyltransferase (ChAT) and AChE activity in the cortex in AD	Cholinesterase inhibitors	Physostigmine, Aricept (donepezil), Exelon (rivastigmine), and Metrifonate	[55,56,57]

**Table 2 ijms-24-14044-t002:** Summary of Nanocarriers Used for AD treatment (only in/ex vivo study mentioned here).

Nanocarrier Material	Modifications/ Functionalized by	Therapeutic Agent	Experimental Model	Route of Administration	Experimental Doses and Periods	Result	Limitations and Possibilities	Reference
Solid Lipid Nanoparticles	Polysorbate 80	Rivastigmine tartrate	Sheep mucosa	Intranasal	0.178 mg/mL 6 months	Formulation shows safe intranasal delivery without any toxicity	An in vivo study is needed to see the safety and effectiveness of formulation	[124]
Superparamagnetic Iron Oxide NPs	-	AβOs Antibody and Class A scavenger receptor activator	Mice	Intravenously	1 mg/day 28 days	Formulation targeting the AβOs improved the uptake of AβOs by microglia	The distribution and the half-life of formulation will be studies in brains for long time benefits	[140]
Carbon Dots	-	Memantine	Wild-type zebrafish	Intravascularly	-	Cross the BBB and inhibes the tau aggregation	This study shows the significant therapeutic potential in AD treatment	[141]
Nano drug particles	-	Drug	AD patients	Oral	- 45 days	Have high stability and less toxicity	A large level study is needed to conclude the role of nanodrug in AD treatment	[142]
Graphene oxide	-	Dauricine	Mice	Intranasal	1 µg/µL 21 days	The formulation protects against oxidative damage and apoptosis	Graphene oxide-dauricine crosses the BBB, enters the brain and shows potential therpeutic agent in AD treatment	[116]
Nanostructured lipid carriers	RBC membrane vesicles	Glycoprotein of Rabies virus and triphenylphosphine cation	APP/PS1 mice	Intravenously	2 mg/kg/every 2 days 30 days	Conquered the ROS-induced mitochondrial dysfunction	Mitochondria-targeted nanosystems showing a capable therapeutic candidate for AD treatment	[143]
Graphene Quantum Dots	-	Flower Clitoria ternatea extract	Rat	-	3 mg/kg 7 days	Improved cognitive behavior and memory capacity in experimental model	QDs significantly crossed the BBB and significantly reduced the Alzheimer-like symptoms in rats; shows potential as therapeutic DDS	[115]
Linoleic acid micelles	-	lactoferrin	Wistar rats	Oral	500 mg/Kg 60 days (AlCl3 induction) and +30 days (treatment regimen)	Enhanced cognitive capabilities, reduced brain OS, apoptosis, inflammation, and AchE activity	Multiple functions of formulation show a greater potential in AD treatment and can be further evaluated at large level	[144]
PLLA/PLGA Hybrid NPs	Hierarchical porous carbon (HPC)	Galantamine	Wistar rats	Intranasal	3 mg/kg 24–48 h (clinical evaluation)	Successful delivery to the hippocampus area	Further evaluation in rodent models of AD is needed	[145]
Amylo Lipid Nanovesicles	Lipid-modified starch hybrid	Curcumin	Rat	Intranasal	160 μg/kg 24 h	Curcumin targeting specifically to the brain	Result shows potential of mylo lipid nanovesicles in drug delivery to brain	[146]
Chiral AuNPs	L- and D-glutathione	NA	KM mice	Intravenous	25 mg/kg 6 days	Formulation crossed the BBB and inhibited Aβ42 aggregation	The formulation shows a promising therapeutic strategy for AD by significantly rescueing the spatial learning and memory impairments in AD model	[147]
PAMAM Dendrimers	Lactoferrin coupled	Memantine	Mice	Intravenous	mg/kg 8 weeks	The formulation enhances the memory improvement	PAMAM dendrimers shows a significant impact on the memory aspects in AD-induced mice and could be further evaluated in other AD model	[148]
SLNPs	-	Erythropoietin	Wistar rats	Injected in CA1 Region of hippocampus	1250 and 2500 U/kg 28 days	Formulation reduced the OS, ADP/ATP ratio and Aβ plaque deposition	The result of in vivo study is encouraging for further evaluations	[149]
Polymeric PLGA NPs	PEG linked	Pioglitazone (PGZ)	Buccal and nasal mucosa	Oral	110 µg/mL 6 h	Cross BBB and drug delivered to the brain	In vivo study is required to see the actual result	[150]
SLNPs	Polysorbate 80, phosphatidylserine or phosphatidic acid	Nicotinamide	Rat	Intravenous and Intraperitoneal	200 mg/kg 4 days	Improving cognition, protecting the neuronal cells and suppress Tau hyperphosphorylation	Not sufficient in improving the AD at early stages	[123]
Polylactide-coglycoside NPs	Lactoferrin-conjugated N-trimethylated chitosan	Huperzine A	KM mice	Intranasal	1 µg/mL 12 h	Have a prolonged release time of the drug and target specific delivery	Evaluation of therapeutic efficacy for formulation in animal AD models is needed	[151]
AuNPs	-	Streptozotocin	Wistar male rats	Intraperitoneal	2.5 mg/kg body weight 21 days	Prevent mitochondrially ATP production, neuroinflammation, and OS	AuNPs treatment prevented the pathological events of brain during AD pathogenesis possibly use for treatment of AD	[152]
PLGA-NPs	PVA	Quercetin	AD control mice	Intravenous	20 mg/kg body Weight 30 days	Significantly improved the spatial memory	The formulation have high therapeutic index and reduced the side effects	[153]
SLNPs	-	Galantamine hydrobromide	Wistar rats	Intraperitoneal	5 mg/kg 26 days	Significant restored memory capability in cognitive deficit Rats	High bioavailability of drug in brain shows potential therapeutic role of formulation	[154]
Nanostructured lipid carrier (NLC)	Lactoferrin modified	Curcumin	Rats	Intraperitoneal	10 mg/kg body Weight 24 h	Penetrate BBB and release the drug in the brain	The in vitro and in vivo results shows the safe and biocompatible nanoformulation and could be used for brain targeted DDS	[155]
SLNPs	-	Rivastigmine	Goat nasal mucosa	Incubated with formulation	10 mg/mL 9 h	Showed intact nasal mucosa with RHT SLN indicating the safety of RHT SLN	In vivo study is needed to observe the actual biodistribution and release profile of drug	[156]
Polyethyleneglycol-poly-caprolactone (mPEG-PCL)	-	Resveratrol	Transgenic Caenorhabditis elegans (C. elegans) (CL4176 strains)	Through culture media	0–100 µg/mL 50 days	The formulation shows protection against OS in C. elegans induced by both γ-ray radiation and amyloid- peptide, confirming the successful development of antioxidant NPs.	Further study in large animals is suggested for confirmation of potential DDS	[157]
Poly(lactic acid) NPs	PEG-coated	B6 peptide (CGHKAKGPRK)	Mice	Intravenous	1 mg/kg 30 days	Showed excellent amelioration in learning impairments, cholinergic disruption	In vivo imaging results show a good biodistribution profile of formulation with significant level of accumulation in the brain	[158]
Liposomes	Modified by adding Cell-penetrating peptide	Rivastigmine	Male rats	Intranasal	1 g/kg body Weight 7 days	Enhance the pharmacological properties of drug and BBB penetration	Liposomes improve the drug delivery to brain by penetrating BBB and decrease the hepatic first pass metabolism and gastrointestinal adversative effects	[159]
Polyethylene glycol-polylactide-polyglycolide NPs	Lectins modified	Fibroblast growth factor	Rat	Intranasal	20 and 40 µg/kg/d 17 days	Significantly improved spatial learning and memory in experimental rats	The formulation has a promising DDS for peptide and protein drugs for CNS and play the therapeutic role in AD	[160]
Albumin NPs	Cyclodextrins	Tacrine hydrochloride	Sheep nasal mucosa/Ex vivo	Nasal delivery	2 mg/mL 24 h	Modified NPs enhanced the drug loading and permeation	In vivo study is required	[161]
Liposomes	Biotin-coated	Aβ-mAb	Post-mortem AD tissue	Incubation	10 pMoles/mg Lipid 30 days	Aβ deposits in post-mortem AD brain	Present study shows the diagnostic potential only	[162]
Poly(n-butylcyanoacrylate) NPs	Coated with polysorbate 80	Rivastigmine	Wistar rats	_	273.0 and 408.2 ng/mL 24 h	Enhanced drug delivery 3.82-fold comparatively without nano-formulations	In vitro release studies were performed; however, in vivo drug release studies were not performed	[163]
Nanoliposomes	-	Metformin	Rats	i.c.v injection	50 mg/kg 21 days	Improve memory deficit and reduce neuroinflammation in streptozotocin-induced AD model	Nanoliposome conjugated metformin enhanced the learning and memory impairment in patients suffering from AD and also suppressed the neuroinflammation	[93]
PEGylated Liposomes	Transferrin	Osthole	APP/PS-1 Mice	Intravenous	10 mg/kg 48 h	The formulation exert a protective effect in animal by targeting the Osthole to brain tissue and accumulating Osthole in the brain for long time of period	The in vivo study along with pharmacodynamic and pharmacokinetic confirmed that liposome-based delivery of osthole has long-term circulation, controlled release, and target-specific release and effects	[164]
PEGylated Liposomes	Glutathione	Anti-amyloid single domain antibody fragment	APPswe/PS1dE9 mice	Intravenous	5 μg/body weight 24 h	Liposome-based delivery of antibody fragments cross the BBB and reach into the brain	Pharmacokinetics and biodistribution studies of formulation confirmed the potential of liposome as DDS	[165]
Liposomes	CPP	ApoE2 Gene	C57BL/6 mice	Intravenous	1 μg pApoE2/g body weight 5 days	Modified liposomes crossed the BBB and delivered the target gene to brain	Liposome-based gene delivery is safe and have potential in AD treatment	[166]
Gold nanospheres and gold nanorods	Polyethylene glycol	CLPFFD peptide	In vitro/SH-SY5Y cells	-	20 μM to 1.4 nM 2 h	Conjugates inhibits Aβ-fibrillation	An in vivo/ex vivo study is needed	[167]
Polyoxometalate Nanospheres	-	Aβ peptide	In vitro/PC12 cells	-	10 μM 48 h	Inhibits the aggregation of amyloid β-peptide in AD model	An in vivo/ex vivo study is needed	[168]
Polymeric micelle	PEG	Curcumin	APPswe/PSEN1dE9 model mice	Intravenous	5 mg/kg 38 weeks	Decreased Aβ plaque accumulation and enhanced cognitive behavior in mice	Formulation shows targeted multiple target strategies that are more effective than single target strategies	[169]
Linoleic acid micelles	-	Lactoferrin	Wistar rats	Orally	AlCl3—100 mg/kg—30 days LF-CLA micelles-500 mg/kg 30 days	Formulation enhanced cognitive capabilities, reduced brain oxidative stress, inflammation	Additional animal studies are essential to reconnoiter the potential effect of formulated micelles in AD prevention	[144]
Polymeric Micelles	Cationic Chitosan	pVGF	C57BL6/J mice	Intranasal	1 mg/kg body weight 7 days	Formulation effectively deliver pVGF gene to the brain	Formulation shows the potential of a nonviral gene delivery system for brain-targeted gene	[170]
Nanomicellizing solid dispersion	-	Curcumin	APPSwe/PS1deE9	Orally	47 mg/kg 17 months	Formulation shows potential therapeutic candidate	This study of the effect on other organ demonstrates that the formulation is safe and have a promising potential as a therapeutic candidate for AD	[171]
Chitosan Nanoparticles	_	Piperine	Wistar rats	Intranasal	0.250 mg/kg/day 22 days	Targated delivery and showing via anti-apoptosis and anti-inflammatory effects	The formulation shows a tremendous efficiency as therpeutic agent, and could be further studied at large level	[172]
Chitosan	-	Hyaluronic acid	SH-SY5Y cells	-	25 µM 48 h	Inhibited Aβ aggregation	An in vivo/ex vivo study is needed to understand the impact of negative and positive surface charges of nano-inhibitors	[173]
Selenium (Se) nanoparticles	Sialic acid	Peptide-B6 peptide	PC12 cells.	-	10 μM 4 h	High permeability and cross the BBB	An in vivo/ex vivo study is needed	[102]
PLGA Nanoparticles	Curcumin	Tet-1 Peptide	GI-1 glioma cells	-	0.75 μg/mL, 1.5 μg/mL and 3 μg/mL 72 h	The formulation destroy amyloid aggregates and exhibit anti-oxidative properties	An in vivo/ex vivo study is needed	[137]
Poly(lactic-co-glycolic)	Pegylated	Dexibuprofen	APPswe/PS1dE9	Oral	39 μg/mL–5290 μg/mL 30 min	It reduces the memory impairment and decrease brain inflammation	The accumulation of drug can be observed on the liver and will be studied for safety purpose	[139]
Fe_3_O_4_@Carbon Dots	Fe_3_O_4_	Curcumin	PC12 cells	-	0.4 mg/mL 24–28 h	Inhibits the extracellular Aβ fibrillation	An in vivo/ex vivo study is needed	[174]

## Data Availability

Not applicable.

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
