# Peer review of "Advancements in the Application of Nanomedicine in Alzheimer’s Disease: A Therapeutic Perspective"

_ijms, 2023, doi:10.3390/ijms241814044_

Round 1

Reviewer 1 Report (Previous Reviewer 2)

In this manu, an overview of the remarkable recent developments in nanomedicine using different kinds of nanoparticles for the management and treatment of AD was provided. However, the overall structure was clear. Additionally, the current review talked about a brief overview of AD, barriers to developing therapeutic agents, the role of nanomaterials, and their clinical aspects, which would be quite helpful for readers.

Major points

1.       The part of Line214-248 seems only related to BBB. So this part is not suitable to be put here since the following parts were classified by nanomaterial types.

2.       In the “4.2. Carbon-Based Nanoparticles” part, only CDs were discussed here. Other carbon-based nanomaterials should be also discussed. For example, graphene, carbon nanotubes.

3.       In the Table2, the experimental periods and doses should be included, which are very important.

 Moderate editing of English language is required.

Author Response

Reviewer 1

Response: We would like to thank you for your careful and thorough reading of this manuscript and for the thoughtful comments and constructive suggestions, which helped to improve the quality of this manuscript. We tried to incorporate all comments in the revised version of the manuscript.

Major points

  1. The part of Line 214-248 seems only related to BBB. So this part is not suitable to be put here since the following parts were classified by nanomaterial types.

Response: We agree with reviewer's advice. Line 214-248 shifted to the discussion part.

  1. In the “4.2. Carbon-Based Nanoparticles” part, only CDs were discussed here. Other carbon-based nanomaterials should be also discussed. For example, graphene, and carbon nanotubes.

Response: We thank the Reviewer for this comment. In agreement, we improved the text in the section. We hope that the information provided will help to improve the readability of the review.

In a recent study by Wang et al., a nanoformulation of graphene oxide (GO)-loaded dauricine was developed for the treatment of AD and was evaluated in an AD experimental model both in vivo and in vitro. The findings demonstrated that GO loaded with dauricine significantly decreased OS by increasing superoxide dismutase levels, lowering reactive oxygen species, and decreasing malondialdehyde levels in vitro. It also significantly improved cognitive memory impairment and brain glial cell activation in mice with Ab1-42–induced AD. Thus, GO-loaded dauricine has the potential to be a useful drug for the rapid treatment of AD. This study showed that GO loaded with dauricine protected against Ab1-42–induced oxidative damage and apoptosis in both AD models [83].

A recent study investigated the role of graphene oxide as a chelator to dissolve the amyloid-β plaques in AD using the density functional theory study. They have determined the chelating ability of various metals and GO by comparing the binding energies between metal–-chelator and metal-Aβ complexes and conclude that GO can be used in future AD drug delivery therapy to target toxic metal–Aβ interactions and reduce Aβ aggregation (https://pubs.rsc.org/en/content/articlehtml/2021/tb/d0tb02985h).

GO may block the mTOR signaling pathway and initiate autophagy in microglia and neurons. When microglia and neurons were co-cultured, GO triggered autophagy in both cell types, particularly in the microglia, which aided in the removal of Aβ and ultimately had a protective impact on the neurons. Additionally, GO was able to decrease the toxicity of Aβ to neurons by its clearance in addition to being non-cytotoxic to microglia and neurons. These findings demonstrate the possibility of GO as an AD treatment (https://www.sciencedirect.com/science/article/abs/pii/S0009279720304749).

Carbon nanotubes (CNTs) have garnered attention for their unique properties and biomedical potential. The ability of high drug loading, and can readily cross the BBB and also facilitate the passage of drugs to the brain via the olfactory route with ease opens up new possibilities for targeted drug delivery and neuroprotection. In recent research, β-cyclodextrin modified CNTs nanocomposite was prepared for delivery of donepezil hydrochloride drug for the treatment of AD and found the significant sensitivity and limit of detection (https://www.sciencedirect.com/science/article/pii/S1452398123010064).

  1. In Table 2, the experimental periods and doses should be included, which are very important.

Response: Thanks for your useful suggestion. In the revised version of the manuscript, we added the experimental periods and doses to the table. We hope that the information provided will help to improve the readability of the review.

Thank you for your significantly helpful review!

Reviewer 2 Report (New Reviewer)

This review illustrates the recent developments of different types of nanoparticles for the management and treatment of Alzheimer’s disease (AD) and gathers relevant information concerning the role of physiological barriers in AD drug design.

 Specific comments. 

- Figure 1. Conditions like obesity, hypertension, high cholesterol, and diabetes that are supposed to increase the risk of developing AD are mentioned in the text. Figure 1 does not provide any new information and can be deleted.

- Figure 4. Nanoparticules used six different access routes to cross the Blood Brain Barrier (BBB). A brief description of each of them and how they contribute specifically to the transport of metallic, carbon lipid, nanoparticles will be quite informative.

 - Food and Drug Administration has approved AD therapies that are currently available but it is surprising to read that none of these medications slow down the disease course or treat AD.

- Why is the Blood–Cerebrospinal Fluid Barrier (BCFB) crucial for the treatment of CNS disorders. How could amino acid transporters facilitate medication delivery to the CNS  

- The possibility that nanotechnology can overcome the limitations due to multidrug resistance is promising but not illustrated with convincing data

- Exosomes are extracellular vesicles that come from the plasma membrane of cells. The role of exosome-based medication delivery has emerged as a  therapeutic carrier for both cancer and neurological disorders. This property has been extensively described in previous reviews and its description in a review focusing on synthetic nanoparticles is inappropriate and repetitive.

Author Response

Reviewer 2

This review illustrates the recent developments of different types of nanoparticles for the management and treatment of Alzheimer’s disease (AD) and gathers relevant information concerning the role of physiological barriers in AD drug design.

Response: We would like to thank you for your careful and thorough reading of this manuscript and for the thoughtful comments and constructive suggestions, which helped to improve the quality of this manuscript. We tried to incorporate all comments in the revised version of the manuscript.

 Specific comments. 

  1. Figure 1. Conditions like obesity, hypertension, high cholesterol, and diabetes that are supposed to increase the risk of developing AD are mentioned in the text. Figure 1 does not provide any new information and can be deleted.

Response: Thanks for your thoughtful advice. Hence, we deleted Figure 2 from the revised MS. And the number of all figures has been changed accordingly in the manuscript.

  1. Figure 4. Nanoparticles used six different access routes to cross the Blood Brain Barrier (BBB). A brief description of each of them and how they contribute specifically to the transport of metallic, carbon lipid, nanoparticles will be quite informative.

Response: We agree with the reviewer's suggestions. In agreement with the reviewer's suggestions, we have added the following important points regarding the different routes to cross BBB.

Paracellular transport is a passive transport process that results in the transport of substances across an epithelium. Tight junctions are the major rate-limiting pathway in the paracellular transport of large molecules across the epithelium (https://www.sciencedirect.com/science/article/abs/pii/B978032352725500006X). Large molecules such as polypeptides are generally excluded from paracellular diffusion due to their hydrophilicity and high molecular mass. Adsorption-mediated transcytosis has attracted much attention because of its potential for large-molecule drug delivery to the brain. Cationic albumin-mediated brain DDS is commonly used however coupling this component with NP DDS is proven to be successful in cross-BBB for CNS delivery. Polymeric NPs are commonly used for adsorption-mediated transcytosis (https://www.sciencedirect.com/science/article/abs/pii/B978012814001700007X). Immunocytes, neutrophils, and lymphocytes have a high degree of mobility and they can cross various barriers to reach the target site and release their medication cargo. In different disorders, cell-mediated transportation is therefore frequently utilized as a DDS. To enhance drug delivery, these cells can be loaded with nanocarriers such as polymeric NPs (https://www.ncbi.nlm.nih.gov/pmc/articles/PMC3062753/). In receptor-mediated DDS, transferrin receptors, low-density lipoprotein receptors, and insulin receptors are commonly used. A complex is formed between the desired drug and a receptor-targeting agent. This component is then linked or incorporated into NPs (https://www.frontiersin.org/articles/10.3389/fnins.2018.01019/full). Transcellular diffusion of molecules is based on their solubility, their molecular mass, and charge (https://www.ncbi.nlm.nih.gov/pmc/articles/PMC2761699/). One important process for drug transportation over the BBB carrier-mediated transcytosis is used for the entrance of nutrition and energy intake into the brain cells. The benefits of carrier-mediated transport have been grafted onto nano DDSs using a technique that involves functionalizing the surface of the tiny molecules with certain ligands (https://www.sciencedirect.com/science/article/abs/pii/B9780128140017000068). Metallic, carbon, and lipid nanoparticles are briefly discussed in sections 4.1, 4.2, and 4.3, respectively.

  1. Food and Drug Administration has approved AD therapies that are currently available but it is surprising to read that none of these medications slow down the disease course or treat AD.

Response: Thanks for your comments on AD therapies. The sentence has been corrected.

“However, they have poor effectiveness in treating AD, demonstrating the necessity for other therapeutic strategies”.

  1. Why is the Blood–Cerebrospinal Fluid Barrier (BCFB) crucial for the treatment of CNS disorders? How could amino acid transporters facilitate medication delivery to the CNS?

Response: Thank you for turning our attention to the role of BCFB and amino acid transporters in the treatment of CNS disorders. In agreement with the reviewer's suggestions, we have added the following important points for the same.

It is possible to achieve widespread drug delivery to the entire brain by opening the BCFB instead of the BBB because the BCSF barrier is relatively leaky compared to the BBB, water-soluble substances that do not cross the BBB do so on the BCSF barrier, and enter CSF at a rate inversely related to molecular weight. (https://www.tandfonline.com/doi/abs/10.1517/17425247.2016.1171315?journalCode=iedd20). Kung Y et al. (2022) effectively validated a strategy for noninvasively and selectively opening the BCFB to promote medication administration into CSF circulation by facilitating drug delivery in the CNS by opening the BCFB barrier with a single low-energy shockwave pulse (https://fluidsbarrierscns.biomedcentral.com/articles/10.1186/s12987-021-00303-x).

A promising prospect for L-type amino acid transporter 1 (LAT1)-mediated brain delivery of CNS medicines is provided by the relatively high expression of the LAT1, which is primarily expressed in the cerebral cortex, blood-brain barrier, and blood-retina barrier. The transporter has proven to be effective in delivering clinically used CNS medications and prodrugs like L-Dopa. The bulk of the time, a parent medication has been conjugated to an amino acid side chain using a biodegradable linker without substituting carboxyl and amino groups in order to effectively bind LAT1 (https://www.ncbi.nlm.nih.gov/pmc/articles/PMC7203094/).

  1. The possibility that nanotechnology can overcome the limitations due to multidrug resistance is promising but not illustrated with convincing data.

Response: Agree with the reviewer's keen observation. In agreement, we improved the text in the section. We hope that the information provided will help to improve the readability of the review.

 “An advanced method for combating MDR in cancer treatment is small interfering RNA (siRNAs) delivered by nanocarriers. This method increases the effectiveness of already available medications by inhibiting the no-flux and efflux-related protein pumps that are engaged in or overexpressed in MDR (https://www.ncbi.nlm.nih.gov/pmc/articles/PMC9571152/; https://www.frontiersin.org/articles/10.3389/fphar.2022.776895/full). This type of nanocarrier-based siRNA technology may be helpful for treating CNS conditions. Simultaneously, the downregulation of the pump involved in drug resistance, combining monotherapy with P-gp inhibitors, targeting P-gp with small molecule inhibitors, and bypassing of pumps related to drug efflux are common nanotechnology methods that could be used to overcome the MDR problem in AD treatment”. 

  1. Exosomes are extracellular vesicles that come from the plasma membrane of cells. The role of exosome-based medication delivery has emerged as a therapeutic carrier for both cancer and neurological disorders. This property has been extensively described in previous reviews and its description in a review focusing on synthetic nanoparticles is inappropriate and repetitive.

 Response: Agree with the reviewer's keen observation and suggestions. The paragraph on exosomes has been deleted from the revised MS.

Thank you for your significantly helpful review!

Round 2

Reviewer 1 Report (Previous Reviewer 2)

Major points

1.In the current manu, the part of Line246-275 is still there. Please delete it.

2. In this new version, the parts of Line 175-188 and Line208-215 were newly added. However, these descriptions did not fit the title” 3. Challenges of Drug Designing for AD Treatment”. In this part, challenges should be the main topic. Some successful cases should not be included in this part.

 Moderate editing of English language is required.

Author Response

The Editor

IJMS

MDPI

Subject: Related to the revised version of the manuscript ijms-2596839.

On behalf of my coauthors, I am pleased to submit a revised version of manuscript ijmss-2596839 entitled “Advancement in the application of nanomedicine in Alzheimer's disease” which considers all the comments from the reviewers. We are grateful for providing us an opportunity to revise the manuscript and hence we have uploaded a detailed point-by-point response to the reviewer's comments (again) indicating all the changes. We also presented our manuscript in the edited version (purple text) so that the reviewer could easily get our corrections. The paper has been proofread a second time by the English language company “Editage” to minimize language issues.

We would like to thank the reviewers for their careful and thorough reading of this manuscript and for the thoughtful comments and constructive suggestions, which helped to improve the quality of this manuscript. I believe that this work will be of considerable interest to investigators working in the field of neurodegenerative diseases.

I would greatly appreciate it if you consider this revised paper for publication. We would be glad to respond to any further questions and comments that you may have.

The paper is not being submitted to any other journal. None of the paper’s contents has been previously published and none of the authors has any conflicts of interest associated with this paper. All authors have read and approved the resubmission of the manuscript. Thank you once again for your valuable review and comments.

Looking forward to hearing soon.

Very respectfully,

Prof. Minseok Song

Dept of Life Science, Yeungnam University, Korea

Reviewer 1

Major points

Response: We would like to thank you for your careful and thorough reading of this manuscript and for the thoughtful comments and constructive suggestions, which helped to improve the quality of this manuscript. We tried to incorporate all comments in the revised version of the manuscript.

  1. In the current manu, the part of Line 246-275 is still there. Please delete it.

Response: Agree with review keen observation. Lines 246-275 were deleted from the revised MS.

  1. In this new version, the parts of Line 175-188 and Line 208- 215 were newly added. However, these descriptions did not fit the title” 3. Challenges of Drug Designing for AD Treatment”. In this part, challenges should be the main topic. Some successful cases should not be included in this part.

Response: Thanks for your thoughtful advice.  In agreement with the reviewer's suggestions, we have shifted these 2 paragraphs in the discussion part.

  1. Moderate editing of the English language is required.

The paper has been proofread by the English language company “Editage” to minimize language issues and the certificate is attached.  

Moreover, if the kind reviewer wants to change more in English language correction kindly provide the paragraph where language correction is needed.

Thank you for your significantly helpful review!

Reviewer 2 Report (New Reviewer)

Author Response

The Editor

IJMS

MDPI

Subject: Related to the revised version of the manuscript ijms-2596839.

On behalf of my coauthors, I am pleased to submit a revised version of manuscript ijmss-2596839 entitled “Advancement in the application of nanomedicine in Alzheimer's disease” which considers all the comments from the reviewers. We are grateful for providing us an opportunity to revise the manuscript and hence we have uploaded a detailed point-by-point response to the reviewer's comments (again) indicating all the changes. We also presented our manuscript in the edited version (purple text) so that the reviewer could easily get our corrections. The paper has been proofread a second time by the English language company “Editage” to minimize language issues.

We would like to thank the reviewers for their careful and thorough reading of this manuscript and for the thoughtful comments and constructive suggestions, which helped to improve the quality of this manuscript. I believe that this work will be of considerable interest to investigators working in the field of neurodegenerative diseases.

I would greatly appreciate it if you consider this revised paper for publication. We would be glad to respond to any further questions and comments that you may have.

The paper is not being submitted to any other journal. None of the paper’s contents has been previously published and none of the authors has any conflicts of interest associated with this paper. All authors have read and approved the resubmission of the manuscript. Thank you once again for your valuable review and comments.

Looking forward to hearing soon.

Very respectfully,

Prof. Minseok Song

Dept of Life Science, Yeungnam University, Korea

  1. Moderate editing of the English language is required.

The paper has been proofread by the English language company “Editage” to minimize language issues and the certificate is attached.  

Moreover, if the kind reviewer wants to change more in English language correction kindly provide the paragraph where language correction is needed.

Thank you for your significantly helpful review!

Reviewer 2

Response: We would like to thank you for your careful and thorough reading of this manuscript and for the thoughtful comments and constructive suggestions, which helped to improve the quality of this manuscript.

This manuscript is a resubmission of an earlier submission. The following is a list of the peer review reports and author responses from that submission.

Round 1

Reviewer 1 Report

The authors have dedicated this review to alzheimer's disease, the barriers to treating it and how Nanomedicine can help create better treatments. Overall there are three major flaws with this review that need to be adressed.

1) The first 6 pages (of 11 where there is text) are so well known and so well covered in the literature it is uneccessary in a current day review (see also point two)

2) of the 150 citations a vast majority of them are recent reviews meaning these topics are already covered often in the literature

3) the major importance of this article is the novel research being done and their results. In the text these were vaguely described or completely skipped. The table is nice and coversa huge amount of literature in a very overviewed way (many of which like polymeric nanoparticles are completely missing in the text). While the table is nice overall it completely misses the importance of the results and what novelty that research brings along with an expert opinion of the results of this research.

Without a more detailed explanation of current researchers etc and an expert opinion of the importance of the results this review is mostly just information that could have been copy and pasted from other reviews already published.  

Minor errors present acceptable for MDPI check

Author Response

The Editor

IJMS

MDPI

Subject: Related to the revised version of the manuscript ijms-2506851

On behalf of my coauthors, I am pleased to submit a revised version of manuscript IJMS- ijms-2506851 entitled “Advancement in the application of nanomedicine in Alzheimer's disease” that takes into account all the comments from the reviewers.

We have also uploaded a detailed point-by-point response to the reviewers indicating all the changes that we have introduced in response to the reviewers. We also presented our manuscript in the edited version (red text) so that the reviewer could easily get our correction. The paper has been proof read by English language company “editage” to minimize the language issues.

We are very thankful for the comments, as they have helped us improved the clarity of the paper and the presentation of the findings. I believe that this work will be of considerable interest to investigators working in the field of neurodegenerative diseases.

I will greatly appreciate it if you consider this revised paper for publication. We would be glad to respond to any further questions and comments that you may have.

The paper is not being submitted to any other journal. None of the paper’s contents has been previously published and none of the authors has any conflicts of interest associated with this paper. All authors have read and approved submission of the manuscript. Thank you once again for your valuable review and comments.

Looking forward to hearing soon.

Very respectfully,

Dr. Nidhi Puranik

Dept of Life Science, Yeungnam University, Korea

Reviewer 1

The authors have dedicated this review to alzheimer's disease, the barriers to treating it, and how Nanomedicine can help create better treatments. Overall there are three major flaws with this review that need to be addressed.

Response: Thank you so much for providing your valuable time in reviewing the manuscript. We tried to incorporate all your comments in the revised version of the manuscript.

  1. The first 6 pages (of 11 where there is text) are so well known and so well covered in the literature that it is unnecessary in a current-day review (see also point two)

Response: Agree with reviewer observation. However, to make an understanding to a layman who will read this review article have to comprehend why there is a need for nanomedicine and why traditional medicine and treatment are failed to cure AD. Therefore, in this review article, we have provided a piece of basic information including AD pathogenesis, challenges- blood-brain barrier, blood-cerebrospinal fluid barrier, and multidrug resistance proteins are introduced.

  1. of the 150 citations a vast majority of them are recent reviews meaning these topics are already covered often in the literature

Response: Agree with the reviewer's keen observation. AD is a progressive neurodegenerative disease that impacted most people worldwide and is one of the hot topics for drug designing as well as for early diagnosis.

This review aims to provide the most recent developments in this promising subject that can offer systematic knowledge to design new initiatives and improve how nanoparticles are used to treat AD.

  1. the major importance of this article is the novel research being done and its results. In the text, these were vaguely described or completely skipped. The table is nice and covers a huge amount of literature in a very overviewed way (many of which like polymeric nanoparticles are completely missing in the text). While the table is nice overall it completely misses the importance of the results and what novelty that research brings along with an expert opinion of the results of this research.

Response: Agree with the reviewer's keen suggestions. We have added some research-based data on polymeric NPS in the text as well as in the table.

Without a more detailed explanation of current researchers etc and an expert opinion of the importance of the results this review is mostly just information that could have been copied and pasted from other reviews already published.  

Response: We have updated the revised file with the importance of the results in Table 2 as well as a current research output as well as expert opinion in this field.

Reviewer 2 Report

In this manu, an overview of the remarkable recent developments in nanomedicine using different kinds of nanoparticles for the management and treatment of AD was provided. However, the overall structure was not so clear and logical. Additionally, the current review talked about too much basic information, but not prominent research progression. So, the current version was not too helpful for readers.

Major points

1.       In Line29-31 of Introduction part, is there any updated data about the people number affected by AD? The data from WHO 2006 are too old for this review.

2.       In the “The Blood-Brain Barrier” part, there were too many basic descriptions about BBB. However, there is no related research progression about nanomedicine applications in AD treatment, which would be the key for this review. In the Line145-151, there was no detailed information.

3.       In Line145-175, this part seems not related to BCFB, so please revise it.

4.       In the “Metallic/Inorganic Nanoparticles” part, only AuNPs were discussed here. That would be stuffless for this review. It suggests adding some results of other representative metallic/inorganic nanoparticles.

5.       In the “9. Carbon-Based Nanoparticles” part, Line290-306 was talking about CDs, while Line307-331 was talking about QDs. However, non-carbon QDs should not be included in this part. In addition, Line290-306 about CDs did not provide enough detailed progression about CDs application in the AD therapy.

6.       In the “10. Lipid-Based Nanocarrier”, why were only SLNPs and dendrimers reviewed? What about the other nanomaterials (liposomes and micelles)?

Minor points

1.       The subtitle of “4. The Blood-Brain Barrier” was suggested to be replaced by “3.1 The Blood-Brain Barrier” and another two subtitles “5. The Blood-Cerebrospinal Fluid Barrier” and “6. The Multidrug Resistance Proteins” need to be revised similarly. Or, just delete these three subtitles since these three parts were involved in “3. Challenges of Drug Designing for AD Treatment”.

2.       Similarly, it’s better to change the number of “8/9/10” subtitles to “7.1/7.2/7.3”.

3.       Line 333-341 was the same as Line348-356. Please revise it.

 Moderate editing of English language is required.

Author Response

The Editor

IJMS

MDPI

Subject: Related to the revised version of the manuscript ijms-2506851

On behalf of my coauthors, I am pleased to submit a revised version of manuscript IJMS- ijms-2506851 entitled “Advancement in the application of nanomedicine in Alzheimer's disease” that takes into account all the comments from the reviewers.

We have also uploaded a detailed point-by-point response to the reviewers indicating all the changes that we have introduced in response to the reviewers. We also presented our manuscript in the edited version (red text) so that the reviewer could easily get our correction. The paper has been proof read by English language company “editage” to minimize the language issues.

We are very thankful for the comments, as they have helped us improved the clarity of the paper and the presentation of the findings. I believe that this work will be of considerable interest to investigators working in the field of neurodegenerative diseases.

I will greatly appreciate it if you consider this revised paper for publication. We would be glad to respond to any further questions and comments that you may have.

The paper is not being submitted to any other journal. None of the paper’s contents has been previously published and none of the authors has any conflicts of interest associated with this paper. All authors have read and approved submission of the manuscript. Thank you once again for your valuable review and comments.

Looking forward to hearing soon.

Very respectfully,

Dr. Nidhi Puranik

Dept of Life Science, Yeungnam University, Korea

Reviewer 2

In this menu, an overview of the remarkable recent developments in nanomedicine using different kinds of nanoparticles for the management and treatment of AD was provided. However, the overall structure was not so clear and logical. Additionally, the current review talked about too much basic information, but not prominent research progression. Therefore, the current version was not too helpful for readers.

Response: Thank you so much for your critical comments and for providing the major and minor points that are required in the manuscript. I fully understand the concern about the basic information; actually, we just focused on the common reader of this paper to comprehend the topics we included in the manuscript. I am sorry to say that it was my view of highlighting the need of discussing nanomedicine and why traditional medicine and treatment are failed to cure AD. Therefore, in this review article, we have provided a piece of basic information including AD pathogenesis, challenges of drug designing- blood-brain barrier, blood-cerebrospinal fluid barrier, and multidrug resistance proteins along with discussing different nanoparticles in context to the challenges in the AD treatment. We have added a few more points that the kind reviewers asked us and hopefully, we reach the expectation of the reviewer. Kindly let us know if the reviewers want to add more topics if necessary, we are happy to do that.

Major points

  1. In Line 29-31 of the Introduction part, is there any updated data about the people number affected by AD? The data from WHO 2006 are too old for this review.

Response: I agree with the reviewer's keen observation. Recent data is updated in the revised MS. Thank you

  1. In the “The Blood-Brain Barrier” part, there were too many basic descriptions about BBB. However, there is no related research progression about nanomedicine applications in AD treatment, which would be the key for this review. In Lines 145-151, there was no detailed information.

Response: I agree with reviewer suggestions. To keep this point in mind, we have added recent research article data showing the role of nanomedicine in crossing BBB. I hope you will like the modification.  

  1. In Line145-175, this part seems not related to BCFB, so please revise it.

Response: Thanks for your nice suggestion. Line 145-1159 is shifted in the 3.1 section.  Lines 160-183 have been deleted from the revised MS.

  1. In the “Metallic/Inorganic Nanoparticles” part, only AuNPs were discussed here. That would be stuffless for this review. It suggests adding some results of other representative metallic/inorganic nanoparticles.

Response: I totally agree with the reviewer thoughts. Therefore, the result of recent research work on other metallic/inorganic NPs was added in the revised MS.

  1. In the “9. Carbon-Based Nanoparticles” part, Line290-306 was talking about CDs, while Line307-331 was talking about QDs. However, non-carbon QDs should not be included in this part. In addition, Line290-306 about CDs did not provide enough detailed progression about CDs application in AD therapy.

Response: Thank you for your valuable suggestion. We have updated the MS by adding research-based data of CDs showing the application in AD treatment in text as well as in table 2.

  1. In the “10. Lipid-Based Nanocarrier”, why were only SLNPs and dendrimers reviewed? What about the other nanomaterials (liposomes and micelles)?

Response: Agree with the reviewer’s keen observation. Recent studies related to the application of liposomes and micelles in AD treatment has been updated in the revised MS (in text as well as in table 2 with reference). I hope you will approve the changes.

Minor points

  1. The subtitle of “4. The Blood-Brain Barrier” was suggested to be replaced by “3.1 The Blood-Brain Barrier” and another two subtitles “5. The Blood-Cerebrospinal Fluid Barrier” and “6. The Multidrug Resistance Proteins” need to be revised similarly. Or, just delete these three subtitles since these three parts were involved in “3. Challenges of Drug Designing for AD Treatment”.

Response: We agree with the reviewer suggestion. Changes are done accordingly. Thank you

  1. Similarly, it’s better to change the number of “8/9/10” subtitles to “7.1/7.2/7.3”.

      Response: Thanks for your thoughtful suggestions. We revised this in the revised version of the manuscript.

  1. Line 333-341 was the same as Line348-356. Please revise it.

Response: Thanks for highlighting the suggestion. I have gone through the paragraph and it looks bit differences. Please suggest us what to do more in this aspect.

Line 333-339: The findings demonstrated that GO loaded with Dau significantly decreased OS by raising superoxide dismutase levels, lowering reactive oxygen species, and decreasing malondialdehyde levels in vitro. It also significantly improved cognitive memory impairments and brain glial cell activation in mice with Ab1-42-induced AD. As a result, GO loaded with Dau has the potential to be a useful drug for the quick treatment of AD. This study has shown that GO loaded with Dau could protect against Ab1-42-induced oxidative damage and apoptosis in both AD models.

Line348-356- We think that SLNP may be an effective method for medications to cross the blood-brain barrier and reach the damaged parts of the central nervous system in people with neurodegenerative diseases like AD and PD [81]. A study shows that nicotinamide-loaded functionalized SLNPs improve cognition in AD animal models by reducing Tau hyperphosphorylation [82]. Rivastigmine tartrate-loaded SLNPs were formulated for enhanced intranasal delivery to the brain for Alzheimer's therapeutics [83]. In a therapeutic study for AD treatment, RVG29-functionalized SLNPs were prepared for the delivery of quercetin to the brain and resulted in a 1.5-fold increase in the drug permeability to the experimental cell line [84].

Comments on the Quality of English Language:

Moderate editing of English language is required.

Response: Thank you so much for pointing on language issues. Therefore, we have sent our manuscript to editage for English proof reading to minimize the errors.

Round 2

Reviewer 1 Report

The authors have made minor changes but the statement still stands. This review is a review of other reviews not of current advances or expert opinion.

The english is at a high quality with minor errors